# Salicylic Acid Improves Growth and Physiological Attributes and Salt Tolerance Differentially in Two Bread Wheat Cultivars

**DOI:** 10.3390/plants11141853

**Published:** 2022-07-15

**Authors:** Neila Abdi, Angeline Van Biljon, Chrisna Steyn, Maryke Tine Labuschagne

**Affiliations:** Department of Plant Science, University of the Free State, Bloemfontein 9301, South Africa; neilaabdi@gmail.com (N.A.); avbiljon@ufs.ac.za (A.V.B.); bothac@ufs.ac.za (C.S.)

**Keywords:** bread wheat, salt stress, salicylic acid, physicochemical traits, yield attributes

## Abstract

Abiotic constraints such as salinity stress reduce cereal production. Salicylic acid is an elicitor of abiotic stress tolerance in plants. The aim of this study was to investigate the effects of salicylic acid on two bread wheat cultivars (SST806 and PAN3497) grown under salt stress (100 and 200 mM NaCl) in the presence and absence of 0.5 mM salicylic acid. The highest salt concentration (200 mM), in both PAN3497 and SST806, increased the days to germination and reduced the coleoptile and radicle dry weights. The shoot dry weight was reduced by 75 and 39%, root dry weight by 73 and 37%, spike number of both by 50%, spike weight by 73 and 54%, grain number by 62 and 15%, grain weight per spike by 80 and 45%, and 1000 grain weight by 9 and 29% for 200 and 100 mM NaCl, respectively. Salicylic acid in combination with 100 mM and 200 mM NaCl increased the shoot, root, and yield attributes. Salicylic acid increased the grain protein content, especially at 200 mM NaCl, and the increase was higher in SST806 than PAN3497. The macro-mineral concentration was markedly increased by an increase of NaCl. This was further increased by salicylic acid treatment for both SST806 and PAN3497. Regarding micro-minerals, Na was increased more than the other minerals in both cultivars. Mn, Zn, Fe, and Cu were increased under 100 mM and 200 Mm of salt, and salicylic acid application increased these elements further in both cultivars. These results suggested that salicylic acid application improved the salt tolerance of these two bread wheat cultivars.

## 1. Introduction

Soil salinity severely constrains crop production worldwide [1]. The total global area of saline and alkaline soils is estimated to be around 830 million hectares, more than 6% of the world’s land [2,3,4,5]. Although the actual cost from lost agricultural production is hard to quantify and varies with crop species and the timing, duration, and severity of the stress, it is apparent that the losses in yield and profit are significant [6]. Yield reductions of 50% in durum wheat under dry land salinity conditions [7], 88% in bread wheat under high irrigation salinity [8], and 70% under alkalinity have been reported [9]. These studies highlight the scale of lost productivity on saline and alkaline soils and the great opportunity if the yields in these environments can be improved.

Salt and drought stress are known to have a significant influence on durum wheat quality. Moreover, salinity affects plant growth and morphology. Seed germination encompasses distinct physiological processes, and the transition from dormancy to germination is not only a critical developmental step in the life cycle of higher plants but also determines the failure or success of the subsequent seedling establishment and plant growth [10]. Seed germination is affected by adverse environmental conditions, including salinity [11]. There is a need to study the effects of this constraint in bread wheat under controlled conditions involving salinity and alkalinity, especially as bread wheat has much higher Na^+^ exclusion than durum wheat. It was reported [12] that salt stress affected wheat production by decreasing the grain yield and increasing the protein content. Drought stress increased mainly the protein content and reduced the thousand-grain weight in a study on 10 durum wheat genotypes under field conditions [13]. A study on salt-tolerant and salt-sensitive varieties grown in lysimeters reported that salinity had a small positive effect on the grain quality of the salt-tolerant genotypes [14], whereas the grain quality of the salt-sensitive genotypes was not affected by salinity [15]. Several control strategies have been proposed and employed to minimize the yield losses and to reduce the effect of this constraint. Such strategies include the selection of appropriate cultivars, crop rotation, changes in soil cultivation, shift of sowing dates, appropriate selection of fertilizers, and plant protection products [16]. In addition to several chemical and biological methods, attention to various natural or synthetic substances that can reduce the impact of abiotic stress factors on crops was increased. When applied to plants, these substances enhance the processes stimulating the plant’s tolerance to biotic and abiotic stresses. For example, the use of salicylic acid has a specificity action in plants to improve the tolerance to biotic and abiotic stresses, because salicylic acid increases phytohormones, which is one of the key factors in plant immunity [17]. The application of salicylic acid activates antioxidant defense responses by stimulating a number of antioxidant enzymes, which are necessary for the protection of plants against osmotic and salt stress, as well as other stresses [18,19]. Salinity stimulates antioxidant enzymes such as superoxide dismutase, peroxidase, catalase, and glutathione reductase activities, while they show a declining trend as a consequence of increasing the salicylic acid level [20].

Salicylic acid plays an important role in plant growth and development due to important physiological roles, such as increasing the plant’s response to stress conditions by increasing the tolerance, as was reported in barley [21], wheat [22], maize [23], and mustard [24] under heavy metal, salinity, drought, chilling, and heat stress, respectively [25]. As an endogenous molecule and a multifaceted plant hormone, salicylic acid regulates physiological and biochemical processes and thereby modulates the plant stress tolerance capacity [26,27]. In addition, it can promote antioxidant reactions [28]. The external application of SA is believed to improve the adaptation to salt and osmotic stress by the generation of reactive oxygen species (ROS) in *Arabidopsis* seedlings [29]. A high concentration of compatible solutes exists primarily in the cytosol, so SA can balance the high concentration of salt outside the cell on one hand and, on the other hand, counteract the high concentrations of sodium and chloride ions in the vacuole [30]. There are many reports on salicylic acid signaling, which found that the salicylic acid-mediated control of important genes is involved in salinity-exposed plants [31,32]. For example, it was reported [33] that SA upregulated the expression of *MYB* and *P5CS* in *D**ianthus*
*superbus*, which facilitated its acclimation to moderate salt stress. The supply of salicylic acid significantly increased the glycine betaine (GB)-mediated tolerance to salinity [34]. This was accompanied by an increase in water potential, the antioxidant system (reduced glutathione (GSH), GSH/GSSG redox state, and glutathione reductase (GR) activity) and decreased Na^+^ and Cl^−^ accumulation, the Na^+^/K^+^ ratio, oxidative stress, and lipid peroxidation [34]. Overall, the salicylic acid-induced accumulation of GB protected photosynthesis and growth against 50 mM NaCl-accrued impacts in *Vigna radiata* through minimizing the accumulation of Na^+^ and Cl^−^ ions, oxidative stress, and maintaining high GSH levels, which led to a reduced cellular redox environment. Other studies also reported on salicylic acid signaling and the salicylic acid-mediated control of important genes involved in salinity-exposed plants [35]. Earlier, the salicylic acid supply was reported to enhance the expression of salt stress-mediated responsive genes encoding sinapyl alcohol dehydrogenase, cinnamyl alcohol dehydrogenase, cytochrome P450, heat shock proteins, and chaperones [36]

Regarding growth and yield, several studies have been done on cotton plants, which showed that salicylic acid (150 mg L^−1^) increased the plant height, number of branches, and total cotton yield compared to the control [36]. It was also noted [37] that salicylic acid improved the root and shoot fresh weights, number of inflorescences, fruit yield, and fruit quality of strawberry cultivars. Similar reports showed that salicylic acid application increased the growth and yield of tomato plants grown in a greenhouse under salt stress [38]. Another study [39] reported that salt stress lowered the chlorophyll content, number of tillers, and K^+^/Na^+^ ratio in wheat, but the application of SA improved all these parameters significantly.

Therefore, the aim of this study was to determine the effect of salicylic acid application on seed germination and seedling growth, physiological parameters, physicochemical composition, and mineral content in two South African bread wheat cultivars grown under salinity stress. The hypothesis is that salicylic acid application could improve the salinity tolerance in these two bread wheat cultivars.

## 2. Results

### 2.1. Salicylic Acid Effects on Germination and Seedling Growth under Salt Stress

The results in Table 1 show a significant variability between the treatments. Compared to the control, the salicylic acid treatment alone caused a significant increase of the germination time of 19% and 20% for PAN3497 and SST886, respectively. The combination of 200 mM NaCl and salicylic acid caused the largest increase in the germination time of 20% for PAN3497 and 15% for SST806. The 200 mM of NaCl negatively affected the germination of seeds (slightly more than 100 mM) in both cultivars. Salicylic acid (T0/SA) and the 200 mM NaCl treatment had a similar effect on germination, and the seeds took longer to germinate (Table 1).

Coleoptile dry weight was significantly reduced by salinity in both cultivars, but the effect was much more severe for PAN3497 (40%) than SST806 (28%) due to the 100 mM NaCl treatment. The addition of SA did not affect coleoptile dry weight for either of the cultivars under the 200 mM NaCl treatment (Table 1). Therefore, salicylic acid had no significant effect on this trait. The same pattern was observed for radicle dry weight, where salicylic acid and 100 mM NaCl caused similar effects, but the 200 mM NaCl treatment had the most severe effect, causing a 45% reduction in PAN3497 and 30% in SST806, with the addition of salicylic acid not making any difference (Table 1).

### 2.2. Plant Growth and Yield Attributes

Salt stress significantly (*p*  ≤  0.01) reduced plant growth in a dose-dependent manner for both the cultivars (Table 2). All analysed characteristics were significantly decreased but to a larger extent for the 200 mM (T2) than the 100 mM NaCl treatment. The 200 mM salt concentration reduced the shoot dry weight by 75% and 39%, root dry weight by 73 and 37%, spike number of both cultivars by 50%, spike weight by 73 and 54%, grain number by 62 and 15%, grain weight per spike by 80 and 45%, and 1000-grain mass by 9 and 29% for PAN3497 and SST806, respectively. However, salicylic acid generally increased the shoot, root, and yield attributes under T1 and T2, except in PAN3497, where the shoot dry weight was not increased by under 100 mM NaCl (Table 2). Under the 200 mM NaCl treatment, the grain mass per SA spike increased by 60% for PAN3497 and 38% for SST806 due to the addition of salicylic acid (Table 2).

Under salinity stress (100 mM and 200 mM), plants treated with 1.5 mM of salicylic acid showed an increase of up to 50% in growth parameters (shoot dry weight and root dry weight) and yield attributes (spike number, spike weight, number of grains per spike, grain weight per spike, and 1000-grain weight). Contrary to the root dry weight, which showed higher increases under 100 mM NaCl, the SA application caused a higher increase in shoot dry weight under 200 mM than 100 mM NaCl. In all, the shoot dry weight, root dry weight, spike weight per plant, grain number per spike, and grain weight per spike were highly affected by salt stress, especially in PAN3497, compared with the control treatment (Table 2).

### 2.3. Effect of S*alicylic Acid* on near Infrared Spectroscopy Measured Characteristics of Two Bread Wheat Cultivars Grown under Salt Stress in the Greenhouse

Salicylic acid generally increased the protein and fiber contents of the seed under salinity stress, but the starch content was decreased (Table 3). Overall, the highest increase of neutral detergent fiber and protein content was found in SST806. Under 200 mM NaCl, PAN3497 had the lowest protein content (47.24% less than the control) (Table 3), while, under the same level of salt, SST806 showed a higher protein content in plants treated with salicylic acid (T2/SA) (Table 3). The starch content of the grains was decreased for both levels of salinity in both cultivars. Compared to the control, the decrease was 17.05% and 24.66%, respectively, for PAN3497 and SST806 in the salicylic acid treatment under salinity stress (T2).

### 2.4. Mineral Composition in Flour of two Bread Wheat Cultivars Grown under Salt Stress and Treated by Salicylic Acid

Under 200 mM NaCl, all measured minerals were significantly increased (Table 4). This increase was higher with the salicylic acid treatment for both SST806 and PAN3497. Under salt stress, Na was increased more than other minerals for both SST806 and PAN3497 by 23% and 78%, respectively. Compared to the control, Mn, Zn, Fe, and Cu were increased under both 100 mM and 200 mM NaCl. Salicylic acid application increased the elements mentioned above under the same conditions for both cultivars. Zn was increased by 26% and 15% for PAN3497 and by 52% and 18% for SST806, respectively, with 100 mM and 200 mM NaCl.

Regarding K^+^, NaCl (100 mM) caused a reduction in K^+^ accumulation in SST806 (47%). However, the salicylic acid treatment significantly (*p* ≤ 0.05) increased K^+^ accumulation by 19% in PAN3497 and 35% in SST806 under 200 mM NaCl (Table 4).

## 3. Discussion

Salinity is a severe environmental stressor that has a negative impact on plant growth and development. It induced a loss of germination and reduced growth and yield attributes of the tested bread wheat cultivars. This was similar to findings of a previous study [40] where salinity treatments caused a decline in the growth and biomass of *B. carinata* cultivars, which might be due to a reduced leaf area, imbalance in plant water status, and low production of photo-assimilates. Seed germination and seedling establishment are the two critical stages in plant growth. These stages are the most sensitive to environmental conditions, including salinity [41]. In plants, high salt levels are associated with ionic stress, which is especially observed with high Na^+^ ion accumulation [42]. It was reported [43] that an increased concentration of important ions like Na^+^ and K^+^ is related to the accumulation of glycine betaine (GB) in wheat plants due to SA application. Additionally, the tolerance to saline environments is associated with osmotic adjustment and with the synthesis of osmoprotectants with the sequestration and accumulation of Na^+^ in the vacuole [44]. Therefore, to decrease the negative effects of salinity stress on seed germination, it is important to know to what extent the genotypic variation in the water uptake pattern during these phases is associated with the salt tolerance of genotypes at the germination stage [45]. Against this abiotic stress, various strategies have been applied either to alleviate or minimize salinity-induced reduction in growth and productivity. Important among these strategies in the application of phytohormones (such as salicylic acid) in order to counteract the consequences of soil salinity [46]. Salicylic acid application improved the process of germination under salt stress. The beneficial role of salicylic acid in the signaling network and in the developmental and adaptation processes of plant species against biotic and abiotic stresses, has long been known. In this study, NaCl (both 100 mM and 200 mM) and salicylic acid separately had significant negative effects on germination and the weights of radicles and coleoptiles. The salicylic acid treatment did not alleviate the negative effects of NaCl in the germination process or the negative effects on the weight of radicles and coleoptiles.

Salicylic acid application significantly improved the growth attributes of the two wheat cultivars, as was also reported in *B. carinata.* This was also observed earlier in maize [47], barley [48], mungbean [49], and mustard [50]. Salicylic acid increased the shoot dry weight, root dry weight, and yield attributes under both 100 mM and 200 mM NaCl. This increase of yield from the application of salicylic acid may be due to the stimulation of its physiological role, such as absorption of ions [51], its role in flowering [52], its role in the photosynthetic processes [53], or increasing the soluble sugar and soluble protein, as was reported in cowpea plants [54], all which can directly or indirectly influence the yield.

The increase of the protein and fiber contents due to SA treatment was higher in SST806 than PAN3497 and was higher for 200 mM NaCl compared to 100 mM NaCl. The protein content increased by 78% for SST806 under the 200 mM NaCl treatment (Table 3). The starch content was reduced incrementally with each treatment from T1 (100 mM NaCl). Therefore, the salicylic treatment contributed to a decrease in the starch content, but this was probably due to the increase of the protein content in the same treatments. The underlying mechanisms of salicylic acid-induced abiotic stress tolerance included that the salicylic acid-mediated accumulation of osmolytes can help maintain osmotic homeostasis and improve the regulation of mineral nutrition uptake [55]. It was indicated [56] that salicylic acid is an endogenous signal molecule for the activation of plant growth and plant defense responses to biotic and abiotic stresses.

The results of this study clearly suggested that SST806 is more salt-tolerant than PAN3497, and an exogenous supply of salicylic acid could alleviate the deleterious effects of moderate salinity on the growth of bread wheat cultivars through enhanced activation of the photosynthetic process and the relief of membrane injury [57]. It was reported [58] that the accumulation of glycine betaine (GB) due to the stimulation of indigenous biomolecules such as salicylic acid, protected the membrane functions and increased activity of the antioxidant system. Its accumulation in plants is beneficial, because it provides a nitrogen source for better root growth and the germination of seeds [59].

The concentrations of Ca, P, and K were lower in flour from wheat grain grown in non-saline soil (control). Salinity stress increased these minerals in the plants (Table 4). This confirmed the results of a previous study [60], where the treatment of mungbean plants with 1.5 mM salicylic acid resulted in a maximum decrease in the concentration of Na^+^ and Cl^−^, while the N, P, K, and Ca concentrations were increased under non-saline and saline conditions. The results from this study suggested that salicylic acid application could alleviate salinity stress and improve the plant growth, yield, and its mineral content.

Under salinity stress, salicylic acid increased the neutral detergent fiber and protein content more in SST806 than in PAN3497. These results are in accordance with previous findings [61], where abiotic stress such as drought caused considerable changes in the grain composition, including a substantial decrease in starch accumulation and an increase in grain protein. It was confirmed [62] that salicylic acid increased the growth and values of important quality characteristics such as the protein content. Overall, the results of this study were in accordance with those of a previous study [63], indicating that salinity decreased the morphological parameters and yield attributes by reducing the functionality of the stomata, reducing photosynthesis and, consequently, reducing plant growth. However, salicylic acid minimized the deleterious effects of salt on the growth and adaptation of plants, which was attributed to the well-developed chloroplasts and high activity of antioxidant enzymes.

## 4. Materials and Methods

The experiment was conducted from September 2020 in the greenhouse at the University of the Free State, South Africa, using two commercial hard red bread wheat cultivars (PAN3497 and SST 806). These cultivars both have a medium growing period. They are the products of two different seed companies. Nothing is known about their root systems, but they are similar morphologically. The growing period was 6 months under temperatures set to 19/25 °C night/day, with a relative humidity of 60–70%. To avoid osmotic shock, the salt was dissolved and added to the water used for irrigation of the plants. Salicylic acid (1.5 mM) was sprayed on the leaves every two days as modified from a method previously described [64], and the different levels of salinity were applied once a week from the tillering stage until the end of the spike formation (physiological maturity).

### 4.1. Soil and Biological Materials

The soil used had a pH of 6.8 and comprised of 50% sand, 10% silt, and 40% clay. The soil was sieved (2 mm) before use. Bread wheat (*Triticumaestivum* L.) cultivars PAN3497 and SST 806 were used. Seeds were sown to pots filled with 2 kg of soil that was analyzed. It contained 7.5 mg kg^−1^ (P), 231.4 mg kg^−1^ (K), 564 mg kg^−1^ (Ca), and 147.6 mg kg^−1^ (Mg).

### 4.2. Measurements

#### 4.2.1. Effects of Salt Stress on Germination and Seedling Growth

In preparation, grains from PAN3497 and SST886 were surface-sterilized with 0.1% sodium hypochlorite solution for 1 min, rinsed with sterile distilled water several times, and briefly blotted with sterile paper towels. The treatments applied were (1): Control—seeds germinated in deionized water, (2): Seeds were germinated in a solution of 100 mM NaCl, (3): Seeds were germinated in a solution of 200 mM NaCl, (4): Seeds were germinated in a solution of 100 mM NaCl with salicylic acid (1.5 mM), and (5): Seeds were germinated in a solution of 200 mM NaCl with salicylic acid (1.5 mM). The experiment was conducted according to a method modified from that described in a previous study [65] and used four replicates of 20 seeds each that were germinated in 12-cm-diameter Petri dishes on Whatman No.1 filter paper at 25 °C ± 2 °C in a growth chamber. The filter papers were changed after 48 h in order to avoid salt accumulation [66]. The emergence of the radical/plumule from the seeds was taken as an index of germination, which was recorded at seven-day intervals. The seedling growth evaluation was carried out as described previously [67].

#### 4.2.2. Morphological Measurements

Seven morphological measurements: shoot dry weight, root dry weight, spike number, spike weight per plant, grain number per plant, grain weight per spike, and 1000-grain mass were done for all treatments at the harvesting stage when plants were totally dry [68].

#### 4.2.3. Physicochemical Composition

Near-infrared spectroscopy (NIR) of bread wheat grains was performed on a Perten NIR (Hägersten, Sweden) model DA 7250. The protein, neutral detergent fiber, and starch contents were measured according to the method described previously [69]. Five replicates were done per sample, and the mean was used for analysis.

#### 4.2.4. Determination of P, Ca, Mg, K, Na, Fe, Mn, Cu, and Zn Contents in Bread Wheat Flour

The mineral extraction was done according to the dry-ashing method [70]. After milling 100 g of grain per cultivar, 2 g of flour for each sample was placed in labeled crucibles and ashed for 3 h in a furnace at a temperature of 550 °C. The samples were allowed to cool before they were digested with 2–2.5 mL of concentrated HNO_3_. Samples were dried in a sand bath before placing them back into the furnace at 550 °C for 1 h. Samples were allowed to cool for 10 min before adding 10 mL of diluted HNO_3_ (HNO_3_:H_2_O 1:2 dilution ratio) and then placed for 5 min in a sand bath. After cooling, the samples were placed into 200-mL volumetric flasks, and milli-Q water was added to fill the volume. The mixture was filtered through Whatman paper for purification. For each of the samples, an aliquot of 50 mL was analyzed for the mineral concentration using the atomic absorption spectroscopy (AAS) method. Five replicates were done per sample.

#### 4.2.5. Statistical Analysis

The experiment was factorial, based on a completely randomized design. The different combinations gave a total of three treatments for each cultivar. The treatments were: control (T0), plants under 100 mM of NaCl (T1), and plants under 200 mM of NaCl (T2). Three replicates were used for each treatment, with 10 pots per replication (three plants per pot), giving a total of 30 plants per replication. ANOVA was performed using the SPSS statistical program v.13 (IBM Corporation, Armonk, NY, USA) (http://oss.software.ibm.com/icu4j/. Accessed 5 April 2022), and subsequent comparison of the means was done using Duncan’s multiple range test at *p* = 0.05. Data are presented as the treatment mean ± SE (*n* = 15) for both growth and yield attributes and (*n* = 5) for the remaining tested characteristics.

## 5. Conclusions

This study demonstrated that the two bread wheat cultivars tested a tolerated salinity, but severe salt stress led to growth inhibition and reduced the plant development, nutrient uptake, and yield attributes. Generally, SST806 was more tolerant than PAN3497 to salt stress. The reason for this is not clear, as these cultivars have similar morphological characteristics, but this should be investigated further. However, salicylic acid minimized the deleterious effect of salt stress on the growth of cultivars. Further research is needed to study the effect of salicylic acid application and GB accumulation in bread wheat cultivars under high saline field conditions.

## Figures and Tables

**Table 1 plants-11-01853-t001:** Effects of salicylic acid under different NaCl concentrations on days to germination, and coleoptile and radicle dry weights of two bread wheat cultivars.

	Days to Germination						
	Treatments						
Cultivars	T0	SA	T1	T2	T1/SA	T2/SA	Cultivar means
PAN3497	1.25 ^d^ ± 0.08	1.55 ^a^ ± 1.03	1.53 ^ab^ ± 0.07	1.56 ^ab^ ± 0.20	1.58 ^a^ ± 1.00	1.56 ^ab^ ± 0.00	1.50
SST806	1.17 ^e^ ± 0.00	1.48 ^bc^ ± 0.05	1.45 ^bc^ ± 0.01	1.47 ^b^ ± 0.00	1.50 ^b^ ± 0.06	1.47 ^b^ ± 0.01	1.43
Means	1.21 ^d^ ± 0.02	1.51 ^ab^ ± 0.9	1.49 ^b^ ± 0.04	1.51 ^ab^ ± 0.43	1.54 ^a^ ± 0.00	1.51 ^ab^ ± 0.27	1.46
	**Coleoptile Dry Weight (g)**						
PAN3497	0.15 ^a^ ± 0.04	0.09 ^b^ ± 0.11	0.09 ^b^ ± 0.06	0.07 ^c^ ± 0.09	0.06 ^d^ ± 0.10	0.07 ^e^ ± 0.00	0.09
SST806	0.18 ^a^ ± 0.12	0.11 ^bc^ ± 0.00	0.13 ^b^ ± 0.00	0.08 ^c^ ± 0.03	0.09 ^c^ ± 0.09	0.08 ^f^ ± 0.34	0.11
Means	0.16 ^a^ ± 0.00	0.10 ^bc^ ± 0.08	0.11 ^b^ ± 0.01	0.07 ^c^ ± 0.08	0.07 ^c^ ± 0.00	0.07 ^e^ ± 0.04	0.10
	**Radicle Dry Weight (g)**						
PAN3497	0.11 ^a^ ± 0.00	0.08 ^b^ ± 0.00	0.08 ^b^ ± 0.00	0.06 ^c^ ± 0.09	0.05 ^c^ ± 0.03	0.06 ^e^ ± 0.00	0.07
SST806	0.13 ^a^ ± 0.18	0.11 ^b^ ± 0.01	0.12 ^ab^ ± 0.02	0.09 ^c^ ± 0.10	0.09 ^c^ ± 0.01	0.09 ^c^ ± 0.12	0.08
Means	0.12 ^a^ ± 0.02	0.09 ^b^ ± 0.07	0.10 ^b^ ± 0.07	0.07 ^c^ ± 0.32	0.07 ^c^ ± 0.00	0.07 ^c^ ± 0.26	0.09

Values in rows followed by different letters are significantly different at *p ≤* 0.05. T0 = control, SA = Salicylic acid, T1 = 100 mM NaCl, T1/SA = 100 mM NaCl + Salicylic acid, T2 = 200 mM NaCl, and T2/SA = 200 mM NaCl + SA. Means ± standard deviation.

**Table 2 plants-11-01853-t002:** The effect of salicylic acid on the yield and yield characteristics of two bread wheat cultivars grown under salt stress under greenhouse conditions.

Cultivar	Treat	Shoot Dry Weight (g)	Root Dry Weight (g)	Spike Number	Spike Weight (g)	Grain Number/Plant	Grain Mass/Spike (g)	1000 Grain Mass (g)
PAN3497	T0	1.23 ± 0.25 ^a^	0.49 ± 0.40 ^bc^	2 ± 0.01 ^a^	2.39 ± 0.10 ^b^	35.33 ± 0.07 ^b^	1.75 ± 0.03 ^b^	51.13 ± 0.00 ^b^
	SA	1.14 ± 0.29 ^a^	0.65 ± 0.02 ^a^	0.66 ± 0.04 ^c^	3.25 ± 0.20 ^a^	46.02 ± 0.12 ^a^	1.94 ± 0.76 ^b^	51.43 ± 0.02 ^b^
	T1	0.73 ± 0.33 ^b^	0.31 ± 0.10 ^c^	0.33 ± 0.07 ^c^	2.1 ± 0.40 ^b^	34.01 ± 0.00 ^b^	1.61 ± 0.03 ^b^	46.16 ± 0.09 ^d^
	T2	0.30 ± 0.04 ^c^	0.13 ± 0.011 ^d^	1 ± 0.00 ^b^	0.63 ± 0.01 ^c^	13.33 ± 0.90 ^d^	0.35 ± 0.87 ^c^	46.66 ± 0.08 ^d^
	T1/SA	0.65 ± 0.08 ^b^	0.60 ± 0.08 ^b^	2 ± 0.00 ^a^	2.79 ± 0.23 ^b^	42.66 ± 0.32 ^ab^	2.52 ± 0.00 ^a^	50.5 ± 0.10 ^c^
	T2/SA	0.50 ± 0.01 ^bc^	0.30 ± 0.17 ^c^	1 ± 0.00 ^b^	0.81 ± 0.38 ^c^	20 ± 0.31 ^c^	0.86 ± 0.10 ^bc^	52.33 ± 0.91 ^a^
SST806	T0	0.88 ± 0.100 ^c^	0.40 ± 0.24 ^c^	2 ± 0.00 ^a^	2.26 ± 0.10 ^b^	35.66 ± 1.20 ^d^	1.77 ± 0.00 ^b^	47.33 ± 1.10 ^a^
	SA	1.32 ± 0.17 ^a^	0.64 ± 0.08 ^b^	2 ± 0.00 ^a^	3.05 ± 0.80 ^a^	49.66 ± 0.10 ^b^	2.41 ± 0.02 ^a^	38.83 ± 0.03 ^d^
	T1	0.67 ± 0.07 ^d^	0.29 ± 0.23 ^d^	0.66 ± 0.10 ^c^	1.63 ± 0.87 ^c^	38.66 ± 0.08 ^d^	1.82 ± 0.95 ^b^	40.5 ± 0.12 ^c^
	T2	0.53 ± 0.03 ^d^	0.25 ± 0.07 ^d^	1 ± 0.00 ^b^	0.97 ± 0.02 ^d^	30 ± 0.59 ^e^	0.97 ± 0.05 ^c^	33.66 ± 0.90 ^e^
	T1/SA	1.09 ± 0.23 ^b^	0.73 ± 0.54 ^a^	2 ± 0.00 ^a^	2.56 ± 0.19 ^b^	57.33 ± 0.45 ^a^	2.48 ± 0.13 ^a^	42.16 ± 0.45 ^b^
	T2/SA	1.37 ± 0.20 ^a^	0.62 ± 0.16 ^b^	0.66 ± 0.80 ^c^	1.90 ± 0.50 ^c^	44.66 ± 0.03 ^c^	1.57 ± 0.16 ^bc^	40.5 ± 0.54 ^c^

T0 = control, SA = salicylic acid, T1 = 100 mM NaCl, T1/SA = 100 mM NaCl + Salicylic acid, T2 = 200 mM NaCl, and T2/SA = 200 mM NaCl + SA. Values in columns followed by different letters are significantly different at *p ≤* 0.05. Means ± standard deviation.

**Table 3 plants-11-01853-t003:** Protein, fiber, and starch contents of two bread wheat cultivars treated with salicylic acid and grown under salt stress in greenhouse conditions.

Cultivars	Treat	Protein Dry Basis %	NDF Dry Basis %	Starch Dry Basis %
PAN3497	T0	16.52 ± 0.00 ^d^	19.55 ± 0.65 ^e^	68.36 ± 0.00 ^a^
	SA	16.62 ± 0.05 ^d^	20.67 ± 0.00 ^d^	68.09 ± 0.00 ^a^
	T1	17.36 ± 0.98 ^c^	19.94 ± 0.00 ^e^	65.78 ± 1.50 ^b^
	T2	17.43 ± 1.80 ^c^	24.0 ± 0.45 ^b^	59.71 ± 0.34 ^d^
	T1/SA	18.85 ± 1.05 ^b^	23.79 ± 0.89 ^c^	63.95 ± 2.00 ^c^
	T2/SA	19.33 ± 0.00 ^a^	26.54 ± 0.34 ^a^	56.7 ± 0.00 ^e^
SST806	T0	15.33 ± 0.04 ^d^	20.58 ± 0.00 ^e^	67.82 ± 1.50 ^a^
	SA	15.46 ± 0.56 ^d^	22.69 ± 1.50 ^d^	65.59 ± 0.85 ^b^
	T1	17.52 ± 0.80 ^c^	20.71 ± 0.90 ^e^	63.16 ± 0.50 ^c^
	T2	21.04 ± 0.00 ^b^	25.96 ± 0.00 ^c^	55.74 ± 1.90 ^e^
	T1/SA	17.52 ± 0.50 ^c^	26.53 ± 2.05 ^b^	59.3 ± 0.50 ^d^
	T2/SA	22.55 ± 0.54 ^a^	30.26 ± 0.00 ^a^	51.14 ± 2.50 ^f^

T0 = control, SA = Salicylic acid, T1= 100 mM NaCl, T1/SA = 100 mM + salicylic acid, T2 = 200 mM, and T2/SA = 200 mM + salicylic acid. Values in columns followed by different letters are significantly different at *p* ≤ 0.05. NDF = neutral detergent fiber. Means ± standard deviation.

**Table 4 plants-11-01853-t004:** Effects of salicylic acid on mineral concentration in flour of two bread wheat cultivars grown under two levels of salinity stress under greenhouse conditions.

Cultivars	Treat	Macro-Minerals (mg kg^−1^)				Micro-Minerals (mg kg^−1^)				
		P	K	Mg	Ca	Na	Mn	Zn	Fe	Cu
PAN3497	T0	719.3 ± 0.0 ^c^	3340.6 ± 0.0 ^b^	974.0 ± 0.0 ^b^	719.3 ± 0.0 ^c^	173.3 ± 0.1 ^d^	14.6 ± 1.0 ^d^	213.4 ± 0.0 ^c^	109.3 ± 0.0 ^cd^	3.8 ± 2.0 ^d^
	SA	782 ± 0.0 ^bc^	3471.3 ± 0.1 ^b^	1241.6 ± 0.0 ^a^	782 ± 0.0 ^bc^	216.6 ± 0.0 ^cd^	28.3 ± 0.0 ^c^	154.9 ± 0.2 ^d^	206 ± 0.00 ^a^	4.4 ± 0.8 ^b^
	T1	895.3 ± 0.1 ^b^	4258.6 ± 0.0 ^a^	1062.6 ± 1.0 ^ab^	895.3 ± 1.1 ^b^	444 ± 1.5 ^b^	61.5 ± 0.2 ^a^	405 ± 0.1 ^b^	89.3 ± 1.1 ^d^	4.0 ± 1.0 ^c^
	T2	900.6 ± 0.9 ^ab^	3424 ± 0.3 ^b^	1036.6 ± 0.5 ^ab^	900.6 ± 0.0 ^ab^	270 ± 0.0 ^c^	61.2 ± 1.0 ^a^	479.8 ± 0.0 ^b^	112.6 ± 0.0 ^cd^	4.4 ± 0.0 ^b^
	T1/SA	1045.3 ± 2.5 ^a^	3557.3 ± 0.0 ^b^	1130.4 ± 0.0 ^a^	1045.3 ± 0.0 ^a^	197.3 ± 1.0 ^d^	58.2 ± 1.5 ^b^	548 ± 0.5 ^a^	88.0 ± 0.9 ^d^	4.5 ± 0.0 ^b^
	T2/SA	940.6 ± 1.5 ^ab^	4228 ± 0.5 ^a^	1125.0 ± 0.1 ^a^	940.6 ± 0.2 ^ab^	1253.3 ± 0.3 ^a^	67 ± 0.0 ^a^	567.0 ± 1.0 ^a^	132.6 ± 0.2 ^b^	6.0 ± 1.0 ^a^
SST806	T0	906.6 ± 0.0 ^d^	6180 ± 0.0 ^a^	1098.2 ± 0.0 ^d^	906.6 ± 0.0 ^e^	193.3 ± 0.3 ^f^	26.2 ± 0.0 ^c^	298.6 ± 0.0 ^d^	83.3 ± 0.5 ^c^	4.6 ± 0.00 ^d^
	SA	802 ± 0.5 ^e^	3104 ± 0.3 ^de^	1175.3 ± 1.5 ^c^	802 ± 1.5 ^f^	306.6 ± 0.5 ^d^	26.4 ± 1.0 ^c^	298.0 ± 0.1 ^d^	151.3 ± 0.0 ^a^	5.2 ± 2.00 ^c^
	T1	995.3 ± 0.1 ^d^	3258.6 ± 0.9 ^d^	1082.6 ± 0.3 ^d^	995.3 ± 0.0 ^d^	544 ± 0.0 ^c^	60.2 ± 0.0 ^b^	319 ± 0.1 ^c^	90.3 ± 0.2 ^c^	4.5 ± 0.50 ^d^
	T2	1056.6 ± 0.9 ^c^	4152.6 ± 0.0 ^c^	1158.7 ± 0.9 ^cd^	1056.6 ± 1.0 ^c^	979.3 ± 1.2 ^b^	75.4 ± 0.9 ^a^	687.0 ± 0.2 ^b^	136 ± 0.2 ^b^	6.6 ± 0.55 ^b^
	T1/SA	1145.3 ± 0.0 ^b^	2557.3 ± 0.1 ^e^	1230.4 ± 0.0 ^b^	1145.3 ± 0.0 ^b^	297.3 ± 0.1 ^e^	58.2 ± 0.0 ^b^	673 ± 0.0 ^b^	86.6 ± 0.0 ^c^	5.5 ± 0.10 ^c^
	T2/SA	1252 ± 0.3 ^a^	6392 ± 0.2 ^b^	1322.6 ± 0.0 ^a^	1252 ± 0.0 ^a^	1282.6 ± 2.0 ^a^	68.0 ± 0.5 ^a^	839.8 ± 0.0 ^a^	156.6 ± 0.8 ^a^	7.9 ± 0.00 ^a^

T0 = control, SA = Salicylic acid, T1 = 100 mM NaCl, T1/SA = 100 mM + Salicylic acid, T2 = 200 mM, and T2/SA = 200 mM + SA. Values in columns followed by different letters are significantly different at *p* ≤ 0.05. Means ± standard deviation.

## Data Availability

The data presented in this study are available at figshare.com (Salicylic acid data.xls).

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
