# Peer review of "Salicylic Acid Improves Growth and Physiological Attributes and Salt Tolerance Differentially in Two Bread Wheat Cultivars"

_plants, 2022, doi:10.3390/plants11141853_

Round 1

Reviewer 1 Report

The paper by Abdi and co-workers describes the effect of SA in salt tolerance in two wheat cultivars. The paper generally describes the response of the two cultivars to SA application under two regimes of NaCl on growth and physiological parameters. The study could be an important addition on the role of SA in salt stress tolerance. There is need of modifying certain parts of the manuscript. The specific comments are below.

-          The title should be changed that gives emphasis to SA in salt tolerance and the parameters. There is no need of specifying the region of the study. The title could be “Salicylic acid improves growth and physiological attributes and salt tolerance differentially in two bread wheat cultivars” or any other with a similar theme.

-          In Introduction, physiological functions of SA have been described, where some ambiguous terms are used, such as ‘specific signaling system’, physiological and biochemical processes’, ‘promote some processes and inhibit others’ and others. Avoid these terms and write specifically the observed parameters.

-          Rewrite the physiological functions of SA in abiotic stress tolerance, particularly salinity stress. In addition to its effects on antioxidant system to promote tolerance, other aspects should also be included.

-          Write uniformly, ‘signalling’ or ‘signaling’.

-          Elaborate the ‘important genes’.

-          How increase in glycine betaine was linked to GR activity?

-          Expand the hypothesis for the study to include the purpose of the study in two bread wheat cultivars, and show that this not a confirmatory study.

-          In results, it is better to use one-decimal values for changes in the observed parameters. It should include the SA concentration used.

-          Compare the effects of 100 and 200 mM NaCl concentration and response of plants grown with these concentrations to SA application.

-          It is written as 200 mM NaCl affected the germination, but not clear it affected the germination positively or adversely. Try to be specific and how SA influenced the parameters in both conditions of NaCl concentrations.

-          The treatments in tables should be arranged as: Control, SA (mM), NaCl (100mM), NaCl (200mM), SA plus 100 mM NaCl and SA plus 200mM NaCl for better understanding.

-          Tables 2 and 3 did not show cultivar or treatment mean. This should be also for Table 1. It is better to present Table 1 data in graph.

-          Sodium concentration increased with SA alone and with SA plus NaCl. Give reasons for this and how SA promoted Na accumulation and then salt tolerance in both the cultivars.

-          Experimental design should be included in the sub head ‘Statistical Analysis”

-          Rewrite Discussion and Conclusion.

-          Discuss the results variation in these two cultivars used under two concentrations of NaCl, and how SA application impacted the observed parameters. This should be followed by the others research on the similar observations or different observations as support. The primary focus of Discussion should be the variation in the observations in the two cultivars under salt stress, and response of these to SA application. On these lines Conclusion will be drawn.

-          Take extra care in formatting the manuscript to the journals’ style.

-          Authors are advised to study the following two papers on SA from 2022 and the references therein to improve the presentation of the manuscript.

-          Journal of Plant Growth Regulation (2022) 41:1966–1977 https://doi.org/10.1007/s00344-021-10565-2

-          Journal of Plant Growth Regulation (2022) 41:1891–1904 https://doi.org/10.1007/s00344-020-10257-3

Author Response

The title should be changed that gives emphasis to SA in salt tolerance and the parameters. There is no need of specifying the region of the study. The title could be “Salicylic acid improves growth and physiological attributes and salt tolerance differentially in two bread wheat cultivars” or any other with a similar theme.

The title was changed  

-     In Introduction, physiological functions of SA have been described, where some ambiguous terms are used, such as ‘specific signaling system’, physiological and biochemical processes’, ‘promote some processes and inhibit others’ and others. Avoid these terms and write specifically the observed parameters.

This was corrected

 -     Rewrite the physiological functions of SA in abiotic stress tolerance, particularly salinity stress. In addition to its effects on antioxidant system to promote tolerance, other aspects should also be included.

This was corrected

 -    Write uniformly, ‘signalling’ or ‘signaling’.

This was corrected

 -     Elaborate the ‘important genes’.

Information was added

 -     How increase in glycine betaine was linked to GR activity?

Information was added

 -     Expand the hypothesis for the study to include the purpose of the study in two bread wheat cultivars, and show that this not a confirmatory study.

This was done

 -     In results, it is better to use one-decimal values for changes in the observed parameters. It should include the SA concentration used.

If we use only one decimal, it becomes difficult to see differences between the values, so we decided to leave it with two decimals

-     Compare the effects of 100 and 200 mMNaCl concentration and response of plants grown with these concentrations to SA application.

This was addressed in the results section

-     It is written as 200 mMNaCl affected the germination, but not clear it affected the germination positively or adversely. Try to be specific and how SA influenced the parameters in both conditions of NaCl concentrations.

This was corrected

-     The treatments in tables should be arranged as: Control, SA (mM), NaCl (100mM), NaCl (200mM), SA plus 100 mMNaCl and SA plus 200mM NaCl for better understanding.

Corrections were made

-     Tables 2 and 3 did not show cultivar or treatment mean. This should be also for Table 1. It is better to present Table 1 data in graph.

We added the means. We tried to do Table 1 as a graph but the results were clearer in the table format, so we decided to stick to a table format.

-    Sodium concentration increased with SA alone and with SA plus NaCl. Give reasons for this and how SA promoted Na accumulation and then salt tolerance in both the cultivars.

An explanation was added

-     Experimental design should be included in the sub head ‘Statistical Analysis”

This was corrected

-     Rewrite Discussion and Conclusion.

This was done.

-     Discuss the results variation in these two cultivars used under two concentrations of NaCl, and how SA application impacted the observed parameters. This should be followed by the others research on the similar observations or different observations as support. The primary focus of Discussion should be the variation in the observations in the two cultivars under salt stress, and response of these to SA application. On these lines Conclusion will be drawn.

The conclusion was rewritten

-     Take extra care in formatting the manuscript to the journals’ style.

This was corrected

Reviewer 2 Report

This seems to be a well-planned and conducted study.  The many claims of differences between cultivars in their responses to the treatments was, in my opinion, not well established statistically.  I would have expected to see ANOVA tests of interactions between cultivar and treatment to justify claims of interactive effects.  Despite using a modified t-test to minimize the danger of multiple comparisons, I am not comfortable with such a reliance upon multiple t-tests to draw conclusions about interactive responses.  I am not a statistician, but I think they should document some justification of the analysis chosen.  

The experiment seems solid, although not especially unique or exciting.

Author Response

This reviewer did not request any revisions.

Reviewer 3 Report

This study reported the effect of SA on two cultivars of wheat under salt stress. The authors found that SA mitigated negative impacts on wheats by salt stress. I listed some points below to improve this manuscript. I hope these comments will help.

Data

Table 1 Please add standard error or standard deviation if there is no particular reason.

Table 2 Please add the units. 

Table 3 Authors compared protein, NDF, and starch at each treatment to something to get %. What are the denominators for these data? The total dry weight of seeds?

Table 4 SA inhibited Na uptake at 100 mM in both varieties. However, SA increased NaCl uptake at 200 mM. Why did SA produce opposite results between 100 mM and 200 mM?

Table 2-4 Are the numbers with ± standard error or standard deviation?

Text

Abstract Line 9            Correct “200 Mm” to “200 mM” 

Page 3, section 2.1.    2.1.1 in “2.1.1. Germination time and seedling growth” is probably unnecessary. Please remove 2.1.1.

Page 3 Line 19           Correct “100 mM NacCl” to “100 mM NaCl”

Page 3 section 2.2 Line 11     Correct “1.5 Mm” to “1.5 mM”

Page 3, section 2.2     Authors wrote, “However, salicylic acid increased the shoot, root and yield attributes under T1 and T2 (Table 2).” But, the Number of T1/SA is smaller than that of T1 in PAN3497 (T1 0.73 vs. T1/SA 0.65). Please describe the data accurately.

Page 4 section 2.3      Only “measured characteristics” in the section title of 2.3 is not italic. 

Page 4 section 2.3 Line1 Correct “fibre” to fiber

Page 4 section 2.3 Line2 Correct Table 4 to Table 3

Author Response

Reviewer 3

Data

Table 1 Please add standard error or standard deviation if there is no particular reason.

Standard deviation was added

Table 2 Please add the units. 

Was corrected

Table 3 Authors compared protein, NDF, and starch at each treatment to something to get %. What are the denominators for these data? The total dry weight of seeds?

Yes, it was total dry weight of seeds

Table 4 SA inhibited Na uptake at 100 mM in both varieties. However, SA increased NaCl uptake at 200 mM. Why did SA produce opposite results between 100 mM and 200 mM?

I think SA affects it positively when salt concentration is more than 100 mM. 

Table 2-4 Are the numbers with ± standard error or standard deviation? 

It is standard deviation

Text

All these following corrections have been made

Abstract Line 9            Correct “200 Mm” to “200 mM” 

Page 3, section 2.1.    2.1.1 in “2.1.1. Germination time and seedling growth” is probably unnecessary. Please remove 2.1.1.

Page 3 Line 19           Correct “100 mMNacCl” to “100 mMNaCl”

Page 3 section 2.2 Line 11     Correct “1.5 Mm” to “1.5 mM”

Page 3, section 2.2     Authors wrote, “However, salicylic acid increased the shoot, root and yield attributes under T1 and T2 (Table 2).” But, the Number of T1/SA is smaller than that of T1 in PAN3497 (T1 0.73 vs. T1/SA 0.65). Please describe the data accurately.

Page 4 section 2.3      Only “measured characteristics” in the section title of 2.3 is not italic. 

Page 4 section 2.3 Line1 Correct “fibre” to fiber

Page 4 section 2.3 Line2 Correct Table 4 to Table 3                                                                                    

Round 2

Reviewer 1 Report

Authors have addressed the concerns satisfactorily.

This manuscript is a resubmission of an earlier submission. The following is a list of the peer review reports and author responses from that submission.

Round 1

Reviewer 1 Report

In this paper, authors have emphasized growth and yield loss due to salinity and subsequent influence on SA on plants grown with two levels of salinity. This is important information on the influence of SA in salinity stress acclimation. The manuscript needs major revision in respect of writing few paragraphs and presentation of data. Last lines of 2nd para (Intro) are not clearly introduced. This needs to be clearly introduced without any ambiguity. Similarly, introducing SA in the last three lines is not appropriate. Authors may consider rewriting the statements as ‘plant growth substances such as SA are known to influence growth and yield of crop plants under optimal and salinity stress conditions.’ The citation [19] may be extended for other crops also, and should not be limited to cotton. SA affects growth and development of several crops including wheat. Based on these available literature on SA, hypothesis of the work should be set with the emphasis what new aspects are introduced in this paper.

Results as written do not show the effects of NaCl/SA or both on the studied parameters. Authors should write clearly how much these parameter were reduced by salinity or increased by SA and the recovery of loss was made by SA in the presence of NaCl. Express the data in % changes. In Table 1, the column with 200 salinity plus SA is missing. There is no need to give ANOVA, and the ANOVA presented in Table 2 is not correct. All tables should be prepared in line with Table 5. This table clearly represents the responses.

Authors should try to expand the discussion to show why the two cultivars responded differentially to NaCl or SA. There are various reports to show effects of SA in presence of salt. These publications show how SA interacted with other biomolecules and plant hormones and maintained essential nutrient levels to reduce NaCl effects. These should be discussed in relation to the two cultivars differential response. Authors may consult following papers to expand the discussion and improve the manuscript.

1.            Control of elevated ion accumulation, oxidative stress and lipid peroxidation with salicylic acid-induced accumulation of glycine betaine in salinity-exposed Vigna radiata L. Applied Biochemistry and Biotechnology 2021, 193: 3301-3320.

2.             The key roles of salicylic acid and sulfur in plant salinity stress tolerance. Journal of Plant Growth Regulation 2020, 1-14.

3.            Salicylic acid increases photosynthesis of drought grown mustard plants effectively with sufficient-N via regulation of ethylene, abscisic acid and nitrogen-use efficiency. Journal of Plant Growth Regulation 2022, doi: 10.1007/s00344-021-10565-2.

4.            Involvement of ethylene in reversal of salt stress by salicylic acid in presence of sulfur in mustard (Brassica juncea L.). Journal of Plant Growth Regulation 2021, doi: 10.1007/s00344-021-10526-9.   

The details of all methods should be provided to enable to repeat the analysis, particularly determination of morphological measurements. Write as ‘Data are presented as treatment Mean ± SE (n = 5).    Conclusion should be based on the effects of NaCl/SA and the responses of the two cultivars

Reviewer 2 Report

Several details are missing in the methodology, compromising results and the replication of this study:

1.       The authors described the experimental procedure as “The experiment was carried out in a greenhouse using two commercial hard red bread wheat cultivars (PAN3497 and SST 806) at the University of the Free State, South Africa. It was a factorial experiment based on a completely randomized design. The different combinations gave a total of three treatments for each cultivar. Treatments were: Control (T0), plants under 100 mM of NaCl (T1) and plants under 200 mM of NaCl (T2). Salt was added to the water used for irrigation. Salicylic acid (1.5 mM) was sprayed on the leaves every two days, and the different levels of salinity were applied once a week from tillering stage until the end of spike formation. Ten replicates were used for each treatment.” I do not follow the factorial experiment and how these gave a total of three treatments for each cultivar. Could you please explain this better?

2.       If the experiment was conducted in a greenhouse (controlled?) what were the other abiotic conditions, e.g, what temperature and type of light were used?

3.       Why was salicylic acid applied every 2 weeks while salt was added only once a week?

4.       I’m sorry but “until the end of spike formation” cannot be followed. Do you mean one month; or two months? Basically, how much did this experience last?

5.       Please give more details concerning the two genotypes used. The methods show no background concerning their specificity to salinity or salinity-tolerance differences (if any).

6.       I cannot follow the experiments that were done directly in the greenhouse than the ones that came from germination (4.2). For instance, did the measurements stated in 4.2.2 come from the germinated seeds or the plants grown in the greenhouse?

7.       About the germination, details are missing concerning how this was done. In what conditions? In what growing chamber (or similar)? How many days did it last? How many seeds were used?

8.       How many plants were used for the morphological measurements, physicochemical composition, and mineral content? The statistical analysis indicates only “Results were the means of five replicates for both growth and yield attributes and three replicates for the remaining tested characteristics”. If so, these are very low numbers and quite below the desired statistical power. The minimum to obtain a test statistic (t or F statistic) is df model + 2. For a t-test that means 1+2 = 3. For an ANOVA with a single explanatory variable with 5 classes that means 4+2. For a regression with two explanatory variables that means 2 + 2. These calculations become more complicated with the several explanatory variables in this study, or with random terms, such as a repeated measure design.

9.       The statistical analysis described is also incomplete. The authors have made an ANOVA between means. Means of what? Judging by the results the authors should have 3 factors: genotypes, salt, and salicylic acid (plus the interactions). Yet, nothing is described, although table 2 shows an ANOVA with 2 factors, cultivar and treatment, and the interaction. However, the reader cannot tell if the treatment is the salinity or the salicylic acid or the two together.

110.   The two cultivars seem to behave differently, at least in terms of growth (Table 2), and some other traits. However, nothing is indicated about that or what could explain these differences. 

The authors have written a very poor discussion with no compelling arguments of why salicylic acid might improve some traits such as growth. It also seems that adding salicylic acid might only have beneficial effects under higher salt concentrations.